# Supplementation with Complex Phytonutrients Enhances Rumen Barrier Function and Growth Performance of Lambs by Regulating Rumen Microbiome and Metabolome

**DOI:** 10.3390/ani15020228

**Published:** 2025-01-16

**Authors:** Juan Du, Yuan Wang, Shaohui Su, Wenwen Wang, Tao Guo, Yuchao Hu, Na Yin, Xiaoping An, Jingwei Qi, Xuan Xu

**Affiliations:** 1College of Animal Science, Inner Mongolia Agricultural University, Hohhot 010018, China; dujuan_ddj@163.com (J.D.); 16604715667@163.com (S.S.); wangwenwen2017@emails.imau.edu.cn (W.W.); guot199608@163.com (T.G.); yuchaohu1994@163.com (Y.H.); yinna0420@163.com (N.Y.); anxiaoping@imau.edu.cn (X.A.); qijingwei@imau.edu.cn (J.Q.); 2Key Laboratory of Smart Animal Husbandry at Universities of Inner Mongolia Autonomous Region, Hohhot 010018, China; 3Integrated Research Platform of Smart Animal Husbandry at Universities of Inner Mongolia Autonomous Region, Hohhot 010018, China; 4Inner Mongolia Herbivorous Livestock Feed Engineering Technology Research Center, Hohhot 010018, China; 5National Center of Technology Innovation for Dairy-Breeding and Production Research Center, Hohhot 010018, China; 6Department of Statistics, Kansas State University, Manhattan, KS 66061, USA; 71DATA Consortium, Kansas State University Olathe, Olathe, KS 66061, USA

**Keywords:** complex phytonutrients, high-concentrate diet, rumen health, apoptosis, inflammation

## Abstract

Complex phytonutrients have attracted extensive interest due to their anti-inflammatory effects. Therefore, in this study, complex phytonutrients were used to intervene in lambs, and it was confirmed that complex phytonutrients enhanced rumen epithelial barrier function, thereby lowering inflammation and inhibiting the overactivation of the JNK/p38 MAPK signaling pathway, optimizing the composition of the rumen microbiota, and increasing the levels of Ursolic acid and other metabolites. The strong associations between rumen bacteria and health-related indicators and differential metabolites were further highlighted.

## 1. Introduction

Ruminant animals are essential for meeting the demands of the increasing global population by providing a variety of high-quality animal products, such as meat, wool, milk, and leather [1]. The rumen is an important special organ in the digestive system of ruminants. It decomposes complex organic matter into volatile fatty acids (VFAs), such as acetic acid, propionic acid, and butyric acid, through the fermentation of feed by its microorganisms, which can be absorbed in large quantities through the rumen wall to meet the energy requirements of ruminants and provide about 70% to 80% of the total metabolic energy for the animal body [2,3]. The rumen epithelium is a significant site of absorption and metabolism of nutrients in ruminants. Additionally, the rumen epithelium also represents an effective barrier between the rumen environment and the host blood circulation. Tight junctions (TJs) situated in the middle layers (the stratum granulosum and the spinosum) act as important components in regulating the permeability of the epithelial barrier function and avoiding toxin translocation [4]. In intensive ruminant farming practices, which aim to enhance growth rates or increase milk yield, animals are often provided with high-concentrate diets. Such diets may cause adverse effects on rumen function, such as lower pH, increased accumulation of volatile fatty acids (VFAs), and decreased fiber degradation, leading to a higher risk of various diseases and metabolic disorders. Furthermore, feeding high-concentrate diets has also been found to affect the rumen microbial community structure and function, as well as its interaction with the host animal, and long-term use can negatively affect ruminant health, production performance, and economic benefits; hence, it has received increasing attention in recent years [5,6]. Implementing dietary adjustments, such as adding phytonutrients, thiamine, and probiotics, has shown encouraging results in boosting animal health by enhancing ruminal microbes and metabolites [7].

Fermented wheat bran polysaccharides (FWBPs) and phenolic phytonutrients have been proposed as potential rumen regulators because of their diverse biological functions, which include anti-inflammatory, antioxidant, and antibacterial properties [8,9,10]. Previous studies have shown that incorporating FWBPs into the milk replacer for lambs affects the rumen bacterial population, promotes rumen development, and improves growth performance [11,12,13]. Additionally, recent findings revealed that supplementing lambs with a mixture of cinnamaldehyde, eugenol, and capsicum oleoresin (CEC) led to improvements in both growth performance and rumen health when they were given a high-concentrate diet [14]. The positive effects of adding FWBPs and CEC to lambs have been demonstrated; however, the mechanism of the impact of the combination of these two rumen regulators on animal growth has not been thoroughly studied.

This study hypothesized that incorporating complex phytonutrients (CPS) into lamb diets could enhance growth performance and promote rumen health by regulating rumen bacteria and metabolites, potentially leading to additive or synergistic effects. These changes were expected to positively impact rumen homeostasis in lambs. As a result, this study aimed to investigate the effects of dietary CPS supplementation on the growth performance, rumen fermentation characteristics, epithelial barrier function, rumen microbiota, and metabolomic profiles of lambs.

## 2. Materials and Methods

### 2.1. Animals, Diets, and Treatments

All procedures related to animal care and use for this study received approval from the Inner Mongolia Agricultural University’s Animal Care and Use Committee with the ethical review approval number [2020069]. This study was conducted at Beichen Co., Ltd. The farm is located in Baotou City, Inner Mongolia, China.

A total of fifty-four Dorper × Small Tail Han female lambs of similar age (3.0 ± 0.0 months) and body weight (30.42 ± 0.54 kg) were selected and divided into 3 treatment groups with 6 replicates of 3 lambs according to the principle of total randomization. The lambs were kept in pens with 3 lambs/pen. The groups received a basal diet (the ratio of concentrate to forage was 75:25) supplemented with 0 (CON), 2.5 (CPS2.5), or 5.0 (CPS5.0) g CPS/kg diet (as-fed basis). The determination of therapeutic doses in this trial was mainly based on the results of our team’s previous study [15]. CPS were mixed with feed ingredients to produce total mixed pellet feed for feeding. The ingredient composition and nutrient levels are shown in Table 1. Using the Association of Official Analytical Chemists (AOAC) methods [16], feed samples were analyzed in duplicate for dry matter (DM; method 934.01), crude protein (CP; method 934.01), ash (method 930.05), calcium (method 927.02), and phosphorus (method 984.27). An ANKOM A2000i fiber analyzer (ANKOM Technology, New York, NY, USA) was used to determine neutral detergent fiber (NDF) and acid detergent fiber (ADF), which were both determined following the method described by Soest et al. [17]. Dietary metabolizable energy was calculated according to the Nutrient Requirements of Meat-Type Sheep and Goats (NY/T816-2021) [18]. All the lambs received feed twice per day (08:00 h and 17:00 h) with free access to fresh water, and the experimental diets were offered ad libitum. Feed consumption and leftovers were recorded every day throughout the study. To accurately assess voluntary feed consumption, the feed quantity given was modified each day to ensure a 5% residue level. The CPS product, whose main components contained fermented bran polysaccharide (≥50%), eugenol (≥0.15%), cinnamaldehyde (≥0.09%), and chili oleoresin (≥0.06%), was a patented product developed by the herbivorous livestock Feed Engineering Technology Research Center of Inner Mongolia Autonomous Region. Each pen (2 m × 2 m) was equipped with separate feeding and watering facilities within a barn that was naturally ventilated and had windows. The feeding experiment continued for a duration of 60 days following a 15-day acclimatization phase.

### 2.2. Sample Collection

On the sixtieth day of the experimental protocol, one lamb randomly selected from each replicate was individually housed in a separate pen and subsequently slaughtered following a 12 h fasting period. A delay of 25 min was maintained between the time of slaughter and the subsequent collection of ruminal fluid. After slaughter, ruminal fluid was collected and filtered through four layers of cheesecloth to assess fermentation parameters. The filtered contents were then transferred to a sterile 2.5 mL centrifuge tube and stored at −80 °C for microbiota and metabolomic analyses. Furthermore, two aliquots of tissue samples were collected from the ventral area of each lamb’s rumen. One sample was placed into a sterile centrifuge tube and stored at −80 °C for various subsequent analyses, such as enzyme-linked immunosorbent assay, quantitative real-time PCR, and Western blotting. The other sample was trimmed to dimensions of 2 cm by 2 cm and submerged in a 4% paraformaldehyde solution for histological examination.

### 2.3. Growth-Performance Estimation and Laboratory Procedures

#### 2.3.1. Growth Performance

All lambs in the pens were weighed on days 0 and 60 before morning feeding, the average daily weight gain (ADG) was calculated, and the provided and leftover feed in each pen (3 lambs) were measured daily to calculate the average daily feed intake (ADFI) of each lamb. In addition, feed samples and leftover feed samples were collected from each group on day 1 and day 60, and DM analysis was performed according to AOAC methods (method 934.01). Subsequently, the dry matter intake (DMI) of the lambs was calculated using the average DM content of feed and leftover feed samples. The feed conversion ratios (FCRs) were calculated as DMI/ADG.

#### 2.3.2. Rumen Morphology Analysis

The evaluation of rumen morphology was conducted following the methodology outlined in Wang et al. [14].

#### 2.3.3. Ruminal Fermentation Parameters

The pH of the ruminal fluid was rapidly assessed using a pH meter (PHS-3C; Shanghai, China). Following this, VFAs were analyzed using gas chromatography (Agilent Technologies 7820A, Santa Clara, CA, USA), in accordance with the procedure outlined by Chen et al. [19]. Additionally, the ammonia nitrogen (NH₃-N) content was measured using the method detailed by Ma et al. [20].

#### 2.3.4. The Concentration of Cytokines

The rumen tissue was treated using a hand-held homogenizer (FA6/10; FLUKO, Shanghai, China) at 4 °C and mixed with an ice-cold 0.9% sodium chloride solution (wt/vol, 1:9). After homogenization, the blend was centrifuged at 4000 rpm for 15 min at 4 °C, and the supernatant was separated for further examination.

The concentrations of interleukin-1β (IL-1β), IL-6, IL-10, IL-12, tumor necrosis factor-α (TNF-α), and interferon-gamma (INF-γ) in rumen tissue were determined using the kit of the Nanjing Jiancheng Bioengineering Institute (Nanjing, China). The protein concentration in the homogenate was measured by employing the Coomassie brilliant blue method, following the instructions outlined by the commercial kits (Jiancheng Bioengineering Institute, Nanjing, China).

#### 2.3.5. RNA Extraction and Real-Time PCR

RNA extraction from rumen tissue samples was conducted utilizing the RNA Simple Total RNA Kit supplied by TaKaRa Biotechnology Co., Ltd., located in Dalian, China. In this study, 2 µg total RNA was utilized to synthesize complementary DNA (cDNA) through reverse transcription, adhering closely to the manufacturer’s guidelines for the TB^®^ Green qPCR method. Primers specific to inflammatory cytokines, tight junction proteins, and the housekeeping gene glyceraldehyde-3-phosphate dehydrogenase (GAPDH) were designed following methodologies established in previous research [14], as detailed in Appendix A. The relative abundance of mRNA corresponding to the target genes was evaluated using the 2^−ΔΔCT^ method, with GAPDH employed as the reference gene for normalization.

#### 2.3.6. Total Protein Extraction and Western Blotting

Proteins were extracted from previously frozen and grated rumen tissue samples using RIPA lysis buffer. The protein concentration was measured with the BCA Protein Assay Kit (Thermo Fisher, Waltham, MA, USA). Following extraction, the proteins were separated via SDS-PAGE and subsequently transferred to 0.45 μm PVDF membranes. To reduce nonspecific binding, the membranes were incubated with 6% skim milk for two hours and then washed with 1×TBST. The membranes were incubated overnight at 4 °C with primary antibodies specific to the target proteins. Subsequently, the secondary antibodies were incubated at room temperature for 1.5 h. The membranes were then washed three times with 1× TBST to remove any unbound antibodies. For normalization of the results, β-actin (Santa Cruz Biotechnology, Dallas, TX, USA) was utilized as the loading control. Protein bands were visualized using the ECL Plus Western Blotting Detection System (ProteinSimple, San Jose, CA, USA), and the intensities of the bands were quantified using ImageJ software (version 1.53K).

#### 2.3.7. Ruminal 16S rDNA Amplification, Sequencing, and Analysis

Genomic DNA was extracted from ruminal fluid samples using an Omega Biotek kit (Norcross, GA, USA) and analyzed through 1.0% agarose gel electrophoresis. Universal primers 338 F and 806 R were employed to amplify the V3–V4 region of the 16S rRNA gene. Following PCR amplification, the resulting products underwent purification, quantification, and homogenization before being subjected to high-throughput sequencing on the Illumina MiSeq platform.

Singletons and chimeric sequences were removed from the dataset, and the remaining tags were classified into operational taxonomic units (OTUs) using UPARSE with a 97% similarity threshold. The representative sequences of the OTUs were annotated against the SSU rRNA database (accessible at http://www.arb-silva.de/, accessed on 18 March 2023). Alpha diversity metrics, including Chao1 and the Abundance-based Coverage Estimator (ACE) for community richness, as well as Shannon and Simpson indices for community diversity, were calculated using publicly available software (Mothur v. 1.30.2; http://www.mothur.org/wiki/Calculators, accessed on 18 March 2023) [21]. For the full dataset, beta diversity analyses were performed utilizing the Bray–Curtis dissimilarity index. These pairwise relationships were visualized through Principal Coordinate Analysis (PCoA), which was conducted using QIIME2 (version 2019.4).

#### 2.3.8. Metabolite Extraction and LC-MS/MS Analysis

Metabolite extraction was performed with 1-2-chlorophenyl alanine and methanol. Quality control was conducted with 20 μL from each sample, while the rest was used for LC-MS analysis. The samples were analyzed in both positive and negative ESI (electrospray ionization) modes. Raw data were processed using Compound Discoverer 3.3 (ThermoFisher) for peak alignment, peak picking, and quantification of metabolites. Metabolite annotation involved the KEGG, HMDB, and LIPIDMaps databases, integrating both positive and negative ionization data for a thorough analysis. Principal component analysis (PCA) and partial least squares discriminant analysis (PLS-DA) were conducted using metaX [22]. Differential metabolites were identified based on criteria of VIP > 1, *p*-value < 0.05, and fold-change thresholds of ≥2 or ≤0.5. Volcano plots filtered metabolites of interest according to log2 (fold change) and −log10 (*p*-value) and were generated using the “ggplot2” package in R. Clustering heat maps were created through z-score normalization utilizing the heatmap package in R. Pathway enrichment analysis was performed using the KEGG database.

#### 2.3.9. Correlation Analysis

Spearman’s rank correlation analysis was conducted using the R language package (version 2.15.3) to evaluate the relationships among different rumen microbial genera (*p* < 0.05 and relative abundance > 0.1%), metabolites in rumen fluid with a VIP > 1.0, and indices related to rumen health (*p* < 0.05).

### 2.4. Statistical Analysis

Continuous variables for growth performance, ruminal morphology, ruminal fermentation parameters, mRNA relative expression, and protein expression were expressed as means, according to the normality of distribution. Among them, the DMI data were collected by averaging the feed intake of 3 lambs in each replicate for statistical analysis, with the other data obtained from the lambs slaughtered in each replicate. One-way analysis of variance (ANOVA) was performed to analyze the normally distributed data, and differences among treatments were evaluated by Duncan’s multiple range test. Meanwhile, orthogonal polynomial analysis was conducted to evaluate the linear and quadratic effects of the increasing levels of CPS on the various indexes. Probability values of *p* < 0.05 (2-tailed) were considered to be statistically significant for all comparisons. For the analysis and visualization of microbiota and metabolome data, R packages, including “heatmap”, “corrplot”, and “ggplot2”, were utilized. Additionally, nonparametric methods were used to compare the differences between the two groups for microbiota and metabolome data by applying the Wilcoxon rank sum test.

All statistical analyses were performed using SAS/STAT software (version 9.2; SAS Institute Inc., Cary, NC, USA), R (version 3.4.3), and the Python (Python 3 version) programming language.

## 3. Results

### 3.1. Growth Performance

As shown in Table 2, CPS supplementation linearly and quadratically increased FBW (*p* = 0.016; *p* = 0.006), ADG (*p* = 0.022; *p* = 0.034), and ADFI (*p* = 0.008; *p* = 0.002). Lambs in the CPS2.5 and CPS5.0 groups had significantly increased FBW (*p* = 0.040), ADG (*p* = 0.046), and DMI (*p* = 0.007) compared with lambs in the CON group. The IBW and FCR of the lambs were not statistically different across the three groups (*p* > 0.05).

### 3.2. Rumen Morphology

The results from Table 3 illustrate that CPS supplementation linearly increased the epithelial cell thickness (*p* = 0.019), with significantly higher values observed in the CPS5.0 group compared to the CON group (*p* = 0.042). In addition, dietary CPS supplementation had no significant effects on the length or width of the rumen papillae, nor on the muscle thickness of the lambs (*p* > 0.05).

### 3.3. Ruminal Fermentation Parameters

The results from Table 4 illustrate that dietary CPS supplementation had no significant effect on rumen pH (*p* > 0.05). CPS supplementation linearly and quadratically reduced NH_3_-N (*p* = 0.004; *p* = 0.002), with significantly lower values observed in the CPS5.0 group compared to the CON group (*p* = 0.036). Additionally, CPS supplementation quadratically increased the proportion of butyrate (*p* = 0.003), with significantly higher values observed in the CPS2.5 and CPS5.0 groups compared to the CON group (*p* = 0.002), with no impact on other ruminal fermentation parameters (*p* > 0.05).

### 3.4. Analysis of mRNA Relative Expression Levels of Tight Junction Proteins and Apoptosis-Related Genes and Cytokine Contents in the Rumen Epithelium

Supplementation with CPS linearly increased *Occludin* expression (*p* = 0.048) and linearly and quadratically increased *Claudin-4* expression (*p* = 0.015; *p* = 0.037). Compared to the CON group, the relative mRNA expression of *Occludin* was significantly elevated in both the CPS2.5 and CPS5.0 groups (*p* < 0.001), and *Claudin-4* expression was significantly higher in the CPS5.0 group (*p* < 0.001). However, there was no statistical difference observed in *ZO-1* and *Claudin-1* levels across the three groups (*p* > 0.05; Figure 1A). Similarly, CPS supplementation linearly and quadratically reduced the relative mRNA expression of *Apaf-1* (*p* < 0.001; *p* < 0.001), *Cyt-C* (*p* < 0.001; *p* < 0.001), *Caspase-3* (*p* = 0.013; *p* = 0.007), *Caspase-7* (*p* < 0.001; *p* < 0.001), *Caspase-8* (*p* = 0.014; *p* = 0.013), and *Caspase-9* (*p* = 0.025; *p* = 0.049). When the lambs were fed the diets supplemented with CPS (both the CPS2.5 and CPS5.0 groups), the mRNA levels of *Apaf-1*, *Cyt-C*, *Caspase-3*, *Caspase-7*, and *Caspase-9* were lower than those of the lambs in the CON group (*p* < 0.001). Meanwhile, no significant difference was observed in the mRNA expression of *Fas*, *Bcl-2*, or *Bax* across the three groups (*p* > 0.05; Figure 1B). Additionally, CPS supplementation linearly and quadratically reduced the IL-1β (*p* < 0.001; *p* = 0.002), IL-2 (*p* < 0.001; *p* = 0.004), and INF-γ (*p* < 0.001; *p* < 0.001) contents. Specifically, compared to the CPS5.0 group, both the CON and CPS2.5 groups had higher IL-1β contents (*p* < 0.05), and the CON group had higher IL-2 contents (*p* < 0.05). INF-γ contents were also significantly decreased in the CPS2.5 and CPS5.0 groups compared to the CON group (*p* < 0.01). No significant differences were noted in the contents of IL-6, IL-10, or TNF-α among the three groups (*p* > 0.05; Figure 1C).

### 3.5. The Protein Expression Levels of JNK/P38 MAPK Signaling Pathways in the Rumen Epithelium

Western blot analysis revealed that CPS supplementation linearly and quadratically increased the protein expression of *Bcl2* (*p* < 0.001; *p* = 0.004) and linearly and quadratically reduced the protein expression of *p65* (*p* = 0.010; *p* = 0.041) and *caspase3* (*p* = 0.007; *p* = 0.004). Specifically, *caspase3* protein levels were significantly decreased in the CPS2.5 and CPS5.0 groups compared to the CON group (*p* < 0.05; Figure 2A), and *p65* protein expression was lower in the CPS5.0 group (*p* < 0.05) than in the CON group. Conversely, *Bcl2* protein levels were higher in the CPS5.0 group relative to the other groups (*p* < 0.05). No significant differences in *Bax* expression were observed across the groups (*p* > 0.05). The results in Figure 2B showed that CPS supplementation linearly and quadratically reduced the protein expression of *JNK* (*p* = 0.038; *p* = 0.005) and *p38* (*p* = 0.023; *p* = 0.014) and linearly reduced the protein expression of *p-JNK* (*p* = 0.019) and *p-p38* (*p* = 0.018). Specifically, both the CPS2.5 and CPS5.0 groups had a lower *JNK* protein expression than the CON group (*p* < 0.05). Also, compared to the CON group, only the CPS5.0 group showed a decrease in *p-p38* level and *p38* expression (*p* < 0.05). However, *p-JNK* levels did not significantly differ among the CON, CPS2.5, and CPS5.0 groups (*p* > 0.05).

### 3.6. Rumen Microbe Diversity and Composition

Based on the above results, we found that dietary supplementation of CPS had positive effects on rumen fermentation parameters, epithelial barrier function, and other indicators, and the effect of CPS5.0 was significantly better than that of CPS2.5. Therefore, we selected the CON group and the CPS5.0 group for microflora and metabolomic analyses in order to reveal the effects of CPS on rumen health and their mechanisms.

Across all 12 samples (CON and CPS5.0, *n* = 6), 1,125,901 raw tags were produced, averaging 93,825 tags per sample. Following size filtering, quality control, and removal of chimeras, 794,934 effective tags were retained, with each ruminal digesta sample yielding 66,244 effective tags. Analysis of alpha diversity indices (Shannon, Simpson, Chao1, and ACE) revealed significant differences (*p* < 0.01) between the CPS5.0 and CON groups (Figure 3A). The beta diversity of rumen microbial communities was assessed using PCoA based on Bray–Curtis distance (Figure 3B), which revealed a clear separation between the CPS5.0 and CON groups, suggesting that CPS5.0 influences both the microbial species and their abundance. A Venn diagram (Figure 3C) depicted the overlap and unique ASVs between the two groups, with a total of 2691 ASVs identified. Further analysis using PCA (Figure 3D) and PLS-DA revealed distinct separation between the sample sets. A random permutation test validated the model’s accuracy, with R2X = 0.407 and R2Y = 0.997 (Figure 3E). At the phylum level, 20 phyla were detected across all samples, while at the genus level, a total of 415 microbial genera were identified. The relative abundances of the 20 most abundant bacterial phyla (A) and genera (B) in the rumen fluids are shown in Figure 4. Among the detected phyla, *Bacteroidota* (39.38% for CON and 43.49% for CPS5.0), *Proteobacteria* (26.19% and 11.91%), and *Firmicutes* (20.75% and 31.22%) were the most dominant, followed by *Fibrobacterota* (1.04% and 2.05%), *Actinobacteriota* (1.08% and 1.53%), and *unidentified_Bacteria* (1.72% and 1.31%). Minor phyla included *Acidobacteriota* (0.23% for CON and 0.05% for CPS5.0), *Desulfobacterota* (0.17% and 0.49%), *Chloroflexi* (0.14% and 0.02%), *Gemmatimonadota* (0.11% and 0.00%), *Verrucomicrobiota* (0.10% and 0.04%), *Cyanobacteria* (0.09% and 0.16%), *Myxococcota* (0.06% and 0.00%), *Gemmatimonadetes* (0.04% and 0.02%), *Euryarchaeota* (0.04% and 0.19%), *Synergistota* (0.04% and 0.12%), *Gracilibacteria* (0.01% and 0.02%), *Nitrospirota* (0.03% and 0.00%), and *Campilobacterota* (0.01% and 0.02%) (Figure 4A). The relative abundances of the major microbial phyla varied between different groups. As shown in Appendix A, CPS5.0 supplementation increased the relative abundances of *Actinobacteriota*, *Desulfobacterota*, and *Firmicutes*, while it reduced the relative abundance of *Proteobacteria*. At the genus level, the predominant genera included *Prevotella* (24.30% for CON and 20.48% for CPS5.0), *Succinivibrio* (10.24% and 4.33%), *Succinivibrionaceae_UCG-001* (11.23% and 4.35%), *Rikenellaceae_RC9_gut_group* (4.52% and 8.94%), *Prevotellaceae_UCG-001* (3.80% and 3.89%), *Succiniclasticum* (3.41% and 3.73%), *Treponema* (2.93% and 2.78%), *Dialister* (1.66% and 1.83%), *Sharpea* (1.80% and 0.57%), *Fibrobacter* (1.04% and 2.05%), *Lactobacillus* (1.16% and 0.21%), *Ruminococcus* (0.71% and 2.23%), *Lachnospiraceae_NK3A20_group* (0.69% and 2.71%), *Olsenella* (0.58% and 1.24%), *Megasphaera* (0.31% and 1.03%), and *Selenomonas* (0.00% and 1.53%) (Figure 4B). The CPS5.0 group increased the relative abundances of *Succiniclasticum*, *Prevotellaceae_UCG-001*, *Rikenellaceae_RC9_gut_group*, *Dialister*, *Fibrobacter*, *Olsenella*, *Lachnospiraceae_NK3A20_group*, *Ruminococcus*, *Megasphaera*, and *Selenomonas*, while it decreased the relative abundances of *Succinivibrio*, *Succinivibrionaceae_UCG-001*, *Prevotella*, *Treponema*, *Sharpea*, and *Lactobacillus* (Appendix A).

### 3.7. Analyses of Correlations Between Ruminal Bacterial and Rumen Health-Related Indices

Correlations between rumen health-related indices and dominant genera (relative abundances greater than 1%) in ruminal samples were assessed (shown in Figure 5). For the growth performance, the relative abundance of *Olsenella* was positively correlated with both ADG (*p* < 0.001) and DMI (*p* < 0.05), while the relative abundance of *Lachnospiraceae_NK3A20_group* was only positively correlated with DMI (*p* < 0.05). Regarding fermentation parameters, the relative abundances of *Megasphaera* and *Dialister* showed positive correlations with butyrate concentrations (*p* < 0.05), whereas *Sharpea* was negatively correlated with butyrate concentrations (*p* < 0.05). For cytokines, the relative abundance of *Succinivibrionaceae_UCG-001* was positively correlated with IL-1β concentrations (*p* < 0.05), whereas *Selenomonas* (*p* < 0.05) and *Ruminococcus* (*p* < 0.01) were negatively correlated with IL-1β concentrations. Meanwhile, both *Succinivibrio* and *Sharpea* showed positive correlations with INF-γ concentrations (*p* < 0.05), whereas *Olsenella* was negatively correlated with INF-γ concentrations (*p* < 0.05). For tight junction proteins, the relative abundances of *Lachnospiraceae_NK3A20_group* (*p* < 0.05) and *Olsenella* (*p* < 0.01) were positively correlated with the mRNA relative expression of *Occludin*. Moreover, the relative abundances of *Rikenellaceae_RC9_gut_group* (*p* < 0.01), *Lachnospiraceae_NK3A20_group* (*p* < 0.01), and *Olsenella* (*p* < 0.05) were positively correlated with the mRNA expression levels of *Claudin-4*. Concerning apoptosis-related genes, the relative abundance of *Olsenella* was negatively correlated with the mRNA expression levels of *Apaf-1* (*p* < 0.05), *Cyt-C* (*p* < 0.05), *Caspase-3* (*p* < 0.01), and *Caspase-7* (*p* < 0.001). Meanwhile, the relative abundance of *Lachnospiraceae_NK3A20_group* was negatively correlated with the mRNA expression levels of *Caspase-7* (*p* < 0.05). In terms of protein expression, the relative abundance of *Sharpea* was positively correlated with the protein expression levels of *JNK* (*p* < 0.05), *Bax* (*p* < 0.01), and *Caspase-3* (*p* < 0.01). Also, the relative abundance of *Succinivibrio* was positively correlated with protein expression levels of *Caspase-3* (*p* < 0.05). However, *Olsenella* showed a positive correlation with protein expression levels of *Bcl-2* (*p* < 0.05) but a negative correlation with the protein expression levels of *Caspase-3* (*p* < 0.05).

### 3.8. Ruminal Metabolic Profiling

We examined the untargeted metabolome profiles of rumen contents to evaluate the differences in rumen compounds between the CON and CPS5.0 groups. LC-MS analysis identified 864 metabolites in the positive ion mode and 506 in the negative ion mode. PCA indicated a nearly complete separation, with slight overlap, between the metabolic profiles of the CON and CPS5.0 groups (Figure 6A). Additionally, the OPLS-DA score plots further illustrated a distinct separation between the groups, and a random permutation test confirmed the model’s high accuracy, yielding R^2^X = 0.926 and R^2^Y = 0.984 (Figure 6B).

Differential metabolite analysis, employing criteria of VIP > 1, *p* < 0.05, and FC > 1.5, identified 34 metabolites (25 in positive mode and 9 in negative mode), consisting of 14 upregulated and 14 downregulated compounds (Figure 7A). Specifically, as detailed in Figure 7B and Appendix A, the CPS5.0 group upregulated the levels of All-Trans-13,14-Dihydroretinol, bicyclo[2.2.2]oct-2-en-1-yl 4-methylbenzene-1-sulfonate, Methionine sulfoxide, 2-acetamido-3-(4-methoxyphenyl)propanoic acid, 2-{[2-(4-methylpiperazino) phenyl] methylene} hydrazine-1-carbothioamide, 2-Amino-1, 3, 4-octadecanetriol, DG O-16:3_28:7, Pilocarpine, 2-Amino-1,3-octadecanediol Ursolic acid, Isorhamnetin, 12-Epileukotriene B4, 2,4-Dihydroxybenzoic acid, LPE O-17:2, Phenylacetylglutamine, Lysope 16:0, and LPE O-16:2. Conversely, it reduced the levels of Chenodeoxycholic acid-3-beta-D-glucuronide, 7-Methylguanosine, lithocholic acid, Naringeninchalcone, (1E,4E)-1,5-bis(4-methoxyphenyl)penta-1,4-dien-3-one, L-Canavanine, 9-Oxo-10(E),12(E)-octadecadienoic acid, Palmitoleic Acid, Prostaglandin K2, 13(S)-HOTrE, MGDG O-8:0_22:4, Indoxylsulfuric acid, N′2-benzylidene-5-hex-1-ynylfuran-2-carbohydrazide, Maslinic acid, 3-Methoxy prostaglandin F1α, 18-β-Glycyrrhetinic acid, and methyl 7-hydroxy-4-oxo-8-propyl-4H-1-benzothiine-2-carboxylate compared to the CON diet.

Logarithmic transformation with a base of 2 was applied to the results to generate the radar chart. Figure 7C illustrates the top 10 metabolites exhibiting the highest difference multiples between the CON and CPS5.0 groups. The results showed that All-Trans-13,14-Dihydroretinol, Ursolic acid, bicyclo[2.2.2]oct-2-en-1-yl 4-methylbenzene-1-sulfonate, Isorhamnetin, 12-Epileukotriene B4, 2-acetamido-3-(4-methoxyphenyl)propanoic acid, 2,4-Dihydroxybenzoic acid, LPE O-17:2, Lysope 16:0, and LPE O-16:2 were prominent for their overall metabolite performance. Pathway analyses indicated that alpha-linolenic acid metabolism was the primary signaling pathway in the KEGG enrichment analysis (Figure 7D).

### 3.9. Analyses of Correlations Between Ruminal Metabolic Profiling and Rumen Health-Related Indices

Spearman’s correlation analysis was conducted to assess the relationship between rumen health indices and the top 10 differential metabolites (VIP > 1, *p* < 0.05, FC > 2). As illustrated in Figure 8, metabolites such as All-Trans-13,14-Dihydroretinol, Ursolic acid, bicyclo[2.2.2]oct-2-en-1-yl 4-methylbenzene-1-sulfonate, Isorhamnetin, 12-Epileukotriene B4, 2-acetamido-3-(4-methoxyphenyl) propanoic acid, and LPE O-17:2 were positively correlated with growth performance, epithelial cell thickness, mRNA relative expression levels of tight junction proteins, and the protein level of Bcl2 (*p* < 0.05). However, these metabolites were negatively correlated with the NH_3_-N concentration, cytokine contents, mRNA relative expression levels of apoptosis-related genes, and the protein expression levels of the JNK/P38 MAPK signaling pathways (*p* < 0.05).

### 3.10. Correlations Between the Bacteria and Ruminal Metabolites

At the genus level, Spearman’s correlation analysis was employed to examine the association between rumen microorganisms with an abundance exceeding 1% and the top 10 differentially expressed metabolites (Figure 9). The strongest positive correlations were observed between the relative abundances of Selenomonas and Ruminococcus and the levels of 12-Epileukotriene B4, 2,4-Dihydroxybenzoic acid, LPE O-17:2, LysoPE 16:0, and LPE O-16:2 (*p* < 0.05). Megasphaera showed a positive association with All-Trans-13,14-Dihydroretinol and 2-acetamido-3-(4-methoxyphenyl) propanoic acid (*p* < 0.05), while Fibrobacter was positively associated with Isorhamnetin (*p* < 0.05). Additionally, significant positive correlations were observed between the relative abundance of Olsenella and the levels of All-Trans-13,14-Dihydroretinol, Ursolic acid, bicyclo[2.2.2]oct-2-en-1-yl 4-methylbenzene-1-sulfonate, and LPE O-17:2 (*p* < 0.05).

## 4. Discussion

### 4.1. Growth Performance and Rumen Fermentation Parameters

In recent years, interest in using phytonutrients and polysaccharides to enhance lamb growth performance, improve ruminal fermentation, adjust the ruminal microbiota, and mitigate stress-related issues has increased [23,24]. Nevertheless, no previous research has explored the synergistic effects of phenolic phytonutrients and polysaccharides from fermented wheat bran on ruminal dynamics and associated processes in lambs. In this study, ADG and DMI were affected by CPS; with increasing levels of added CPS, ADG and DMI increased in a quadratic or linear manner, and body weight increased at the end of the experiment, indicating that the moderate inclusion of CPS in the diet of the lambs promoted feed intake and growth, which is consistent with the results of feeding lambs grape-seed proanthocyanidins [25]. The improvement in DMI was attributed to the appealing flavor and taste of CPS, with phenolic compounds in CPS (notably, certain prominent CEC) likely enhancing feed palatability and, in addition, resulting in a higher nutrient intake that would partially explain the higher ADG observed. Moreover, FCR gradually decreased with the increase in CPS addition, but the decrease was not statistically significant, which may have been due to the higher DMI in the CPS group. Although other studies have shown that phytonutrients increase weight gain in sheep but also the efficiency of energy use [26,27], in the current experiment, apparently, rumen fermentation was not the main way in which growth was improved, though rumen health could explain the improvements observed here. The rumen plays a crucial role in the growth, development, and performance enhancement of ruminants. A pivotal component of rumen development is the growth of the rumen epithelium, which supports various physiological functions, such as nutrient absorption, metabolite transport, metabolic activity, and protection [28]. These factors directly influence ruminant production performance. Various morphological indices provide insights into the functionality of the rumen, including the measurement of rumen papillae dimensions, which play a vital role in nutrient absorption, as well as the thickness of muscle layers and epithelial cells, which are essential for maintaining structural integrity and facilitating digestive processes [28]. The results from our study revealed that rumen epithelial cell thickness increased linearly after CPS addition in lambs, which is a key factor in enhancing rumen absorptive capacity, consistent with the findings of Wang et al. [14]. The concentration of NH₃-N in the rumen serves as a crucial indicator of the metabolic balance between dietary protein breakdown and microbial protein utilization. Our results are consistent with those of Cardozo et al. [29] and Geraci et al. [30], who also reported decreased ruminal ammonia levels associated with similar CPS supplementation. Additionally, the role of butyrate in rumen health and development has gained significant attention in recent years. Butyrate, a short-chain fatty acid, functions not only as an energy source for epithelial cells but also significantly contributes to boosting the expression of genes associated with VFA absorption in the ruminal epithelium. This stimulation of gene expression facilitates cellular proliferation and contributes to the growth and development of rumen papillae, which are essential for maximizing nutrient absorption and overall animal performance [31,32]. Therefore, the quadratically increased butyrate concentrations observed in lambs supplemented with CPS may partially explain the enhanced ADG, DMI, and epithelial cell thickness noted in this study.

### 4.2. Rumen Epithelial Barrier Function

In ruminants, a high-concentration diet can temporarily boost growth and milk yield. However, it may also increase volatile fatty acid production, resulting in a lower rumen pH and disruption of the rumen epithelial barrier [33,34]. Such disturbances may result in the translocation of LPSs, which could trigger systemic inflammation and disease, ultimately leading to economic losses [35,36]. In the current study, the integrity of the rumen epithelial barrier in lambs from the CPS group was observed to improve significantly. Key tight junction proteins, such as *ZO-1*, *Occludin*, *Claudin-1*, and *Claudin-4*, are essential for maintaining the structure and function of the ruminal intercellular barrier [37,38]. Research has shown that bioactive compounds derived from natural plants significantly enhance the integrity and functionality of the gastrointestinal tract by modulating the expression of tight junction proteins and associated cytokines [39]. Furthermore, Mu et al. [25] discovered that adding grape-seed anthocyanins, which are abundant in polyphenols, to the diet increased the expression of *Claudin-1* protein in the colonic epithelium, particularly in comparison to lambs fed a high-concentration diet. The results of this study showed that 5.0 g/kg CPS dietary supplementation elevated tight junction proteins, specifically *Occludin* and *Claudin-4* gene expression, in rumen epithelial tissue, which linearly and quadratically increased with the increase in CPS level. These indicated a significant improvement in the integrity of the rumen epithelial barrier among lambs given this dietary supplement.

Dai et al. [40] reported that long-term feeding of high-concentrate diets can induce excessive apoptosis in the rumen epithelium. Apoptosis is crucial for homeostasis, immune response, and development and can be classified into two pathways: extrinsic (receptor-mediated) and intrinsic (mitochondrial) [41]. The extrinsic apoptotic pathway is triggered by external signals through the Fas receptor, activating proteins such as *Caspase-8* and *Caspase-3/7*. In contrast, the intrinsic apoptotic pathway begins with assembling a cytochrome c complex. This process involves *Apaf-1*, *caspase-9*, and dATP, culminating in activating additional caspases, including *Caspase-3* and *Caspase-7* [42]. Our research demonstrated that the supplementation with CPS at doses of 2.5 or 5.0 g/kg significantly diminished apoptosis in rumen epithelial cells. This reduction occurred through a decrease in the mRNA expression of critical components within the apoptotic mitochondrial pathway, specifically *Apaf-1*, cytochrome C (*Cyt-C*), *Caspase-9*, *Caspase-3*, and *Caspase-7*. Furthermore, the regulation of apoptosis is influenced by the *Bcl-2* family of proteins, which encompasses both pro-apoptotic members (such as *Bax*) and anti-apoptotic members (such as *Bcl-2*), along with the activation of *caspase-3* [43]. Western blot analyses revealed that incorporating 5.0 g/kg CPS into the diets of lambs led to a linearly and quadratically reduced protein level of *Caspase-3* while simultaneously elevating the levels of *Bcl-2*. These results highlight the potential of dietary CPS to modulate apoptotic pathways in rumen epithelial cells.

IL-1β, IL-12, and INF-γ are important inflammatory mediators, and elevated levels suggest the presence of inflammation. Our study found that the addition of high levels of CPS (5.0 g/kg) to the diet reduced the concentrations of pro-inflammatory cytokines IL-1β, IL-12, and INF-γ and that they decreased in a linear and quadratic curve as CPS levels increased, suggesting that CPS supplementation could reduce the inflammatory response in lambs. This aligns with the findings of Wang et al. [14], who reported that the addition of cinnamaldehyde, eugenol, and capsicum oleoresin suppressed the cytokine expressions of TNF-α, IL-6, and IL-1β in lambs. Moreover, grape-seed procyanidin supplementation also reduced IL-1β, IL-2, IL-6, and TNF-α levels in colonic tissue of lambs fed a high-concentrate diet [25]. Of particular note, it has been reported that both FWBPs and CEC possess strong antibacterial and anti-inflammatory properties, which may be key to CPS’ ability to alleviate inflammation in lambs. To explore the molecular mechanisms underlying CPS’ effects on rumen inflammation, we examined the expression of key proteins involved in the JNK/p38 MAPK signaling pathways. MAPK signaling is crucial for regulating cell survival, proliferation, differentiation, and apoptosis [44]. In mammals, MAPKs are categorized into three primary groups: extracellular signal-regulated kinase (ERK1/2), C-Jun N-terminal kinase (JNK), and p38 MAPK [45]. Our findings demonstrate that supplementation with 5.0 g/kg CPS significantly reduced the protein levels of JNK, p38, and phosphorylated p38 (P-P38) in the rumen epithelia of lambs and that they decreased in a linear and quadratic curve with the increase in CPS levels. This suggested that dietary CPS may mitigate rumen inflammatory responses by inhibiting the JNK/p38 MAPK pathways, decreasing apoptosis linked to high-concentrate diets, and improving the functionality of the rumen barrier.

### 4.3. Rumen Microflora and Metabolomic Analysis

The rumen microbiota plays a crucial role in maintaining rumen health. To elucidate how CPS supplementation affects rumen bacteria in lambs, 16S rRNA sequencing was conducted. The findings indicated that the inclusion of CPS supplements led to an increase in the diversity of rumen bacteria, which was quantitatively supported by a rise in the Shannon diversity index. This enhancement in microbial diversity is critical, as a more varied bacterial population is associated with improved resilience against disturbances and better nutrient utilization. Furthermore, PCoA, PCA, and OPLS-DA analyses demonstrated notable changes in the bacterial community structure, indicating that CPS supplementation not only enriched microbial diversity but also modified the relative abundances of specific bacterial taxa. Consistent with previous studies on sheep [46], the predominant phyla identified in this experiment were Bacteroidetes, Firmicutes, and Proteobacteria. Bacteroidetes in particular contribute to breaking down a variety of complex polysaccharides, which in turn improves nutrient absorption [47], while Firmicutes are crucial for fiber and cellulose degradation [48]. The 5.0 g/kg CPS supplementation significantly increased the relative abundances of both Bacteroidetes and Firmicutes, consistent with the findings from De et al. [49] that suggested that dietary CPS promote the breakdown of plant fibers and polysaccharides in the rumen. Conversely, the relative abundance of Proteobacteria was significantly reduced, suggesting a potential boost to animal immunity, since this phylum includes pathogens such as *Escherichia coli* and *Salmonella*, which are linked to gastrointestinal infections and inflammation [50]. Importantly, CPS supplementation increased the presence of beneficial bacteria like *Fibrobacter*, *Ruminococcus*, *Succiniclasticum*, *Rikenellaceae_RC9_gut_group*, *Dialister*, *Lachnospiraceae_NK3A20_group*, *Megasphaera*, and *Selenomonas*. These bacteria function as natural probiotics that support rumen health in livestock [51,52,53]. Among them, members of the *Ruminococcus* family are key producers of short-chain fatty acids and promote anti-inflammatory regulatory T-cell differentiation [54]. Our correlation analysis indicated a negative association between *Ruminococcus* abundance and IL-1β levels. A number of studies have shown that *Rikenellaceae_RC9_gut_group* has a high relative abundance in animals with superior growth and health outcomes [55,56]. This genus also increased significantly in our study and positively correlated with claudin-4 levels. This finding parallels results from research on Naringin (a bioflavonoid), which also elevated *Rikenellaceae_RC9_gut_group* in goats fed a high-concentrate diet [57]. Butyrate, primarily produced by *Firmicutes phylum* [58], was positively correlated with the relative abundances of *Dialister* and *Megasphaera* in the present study. The *Lachnospiraceae* family is crucial for breaking down plant cellulose and polysaccharides into volatile fatty acids like acetate, butyrate, and propionate [59]. In this research, the abundance of *Lachnospiraceae_NK3A20_group* increased significantly and was positively correlated with Occludin and claudin-4 levels. This may explain the higher butyrate concentrations observed in the CPS5.0 group, as Huang et al. [60] also found that the *Lachnospiraceae NK3A20* group promotes rumen development by enhancing butyrate production. Additionally, *Olsenella*, a beneficial actinomycete, showed a positive correlation with growth performance, tight junction proteins, and Bcl-2, while negatively correlating with Apaf-1, Cyt-C, Caspase-3, and Caspase-7. These findings suggest that the growth-promoting and immune-enhancing effects of CPS may be linked to the modulation of *Olsenella* abundance.

The rumen microbiota plays a crucial role in producing various metabolites critical for the health and development of lambs, with changes in these metabolites influencing both rumen function and overall host health [61]. In this study, metabolomic analysis highlighted significant differences in the rumen metabolite profile between the CPS5.0 and CON groups regarding amino acids, lipids, fatty acids, cofactors, and vitamins. These findings underscore how dietary interventions can influence metabolite composition and potentially affect rumen and host health outcomes. Several metabolites, including All-Trans-13,14-Dihydroretinol, Ursolic acid, bicyclo[2.2.2]oct-2-en-1-yl 4-methylbenzene-1-sulfonate, Isorhamnetin, 12-Epileukotriene B4, 2-acetamido-3-(4-methoxyphenyl)propanoic acid, 2,4-Dihydroxybenzoic acid, LPE O-17:2, Lysope 16:0, and LPE O-16:2, were significantly upregulated. Recent research highlights the role of microbial metabolites of retinol in maintaining intestinal immune homeostasis [62]. Retinoic acid, a microbial metabolite of retinol, helps maintain intestinal balance by suppressing INF-γ production and regulating IgA levels [63]. Our study found that all-trans-13, 14-dihydroretinol, an intermediate in the conversion of retinol to retinoic acid [64], was elevated in the CPS5.0 group. Moreover, ursolic acid, known to boost the cellular immune response and influence intestinal microbial diversity, also exhibits antimicrobial, anti-inflammatory, and antioxidant properties [65]. 2,4-Dihydroxybenzoic acid has demonstrated notable antimicrobial effects against pathogens, such as *Salmonella*, *E. coli*, and *L. monocytogenes*, with Kalinowska et al. [66] reporting its potent antimicrobial activity against *P. aeruginosa* and *C. albicans*. Isorhamnetin, a significant flavonoid class, has broad pharmacological properties [67], including anti-inflammatory effects through the inhibition of p38 MAPK, ERK 1/2, and JNK phosphorylation [68], as well as reduction in inflammatory cytokines. These observations align with our study, suggesting that these metabolites may support rumen health.

Furthermore, lipid metabolism within the rumen is very active [69], and plant polyphenols and flavonoids significantly regulate animal lipid metabolism by influencing gastrointestinal microbial composition, activating energy receptors, modulating mitochondrial biosynthesis, and mitigating lipid oxidative stress. These actions enhance growth performance and overall animal health [70]. Phospholipids, which provide essential structural support and surface hydrophobicity, are crucial in maintaining enterocyte monolayer permeability and vital for nutrient absorption and waste elimination [71]. In this study, phospholipids, such as LPE O-17:2, Lysope 16:0, and LPE O-16:2, were significantly increased following the administration of CPS, which are rich in polyphenols. Based on our correlation analysis results, it was determined that these metabolites may reduce inflammation and enhance rumen health through lipid metabolism.

Further KEGG functional analysis revealed that alpha-linolenic acid metabolism, involving a polyunsaturated fatty acid with anti-inflammatory and antioxidant properties, played a key role in regulating intestinal flora. Consequently, CPS exhibit substantial rumen-protective potential, with lipids, predominantly phospholipids and polyunsaturated fatty acids, being important contributors. Previous studies have shown a strong link between changes in rumen microbial populations and metabolite levels [46], which is consistent with our results. We noted a positive association between the abundances of *Selenomonas* and *Ruminococcus* and metabolites such as 12-Epileukotriene B4, 2,4-Dihydroxybenzoic acid, LPE O-17:2, Lysope 16:0, and LPE O-16:2. This suggests that there may be reciprocal regulation between these metabolites and microbial communities.

## 5. Conclusions

In summary, these results indicate that dietary supplementation with CPS can reduce rumen tissue inflammatory cytokines and rumen epithelial cell apoptosis, increase the expression of tight junction protein genes, and possibly inhibit the activation of the JNK/P38 MAPK inflammatory pathway by altering the composition and metabolite profiles of the ruminal microbiome, thereby enhancing rumen epithelial barrier function, improving rumen health, and improving lamb performance. Under our experimental conditions, the optimal dose of CPS was 5.0 g/kg of diet. Overall, incorporating CPS into the diet could serve as a promising dietary rumen enhancer to alleviate the adverse impacts associated with high-concentrate feeding in intensive ruminant systems. The microbiota and metabolites were significantly affected by high doses of CPS, and there is merit in further investigation to explore the potential of using them as biomarkers for rumen health.

## Figures and Tables

**Figure 1 animals-15-00228-f001:**
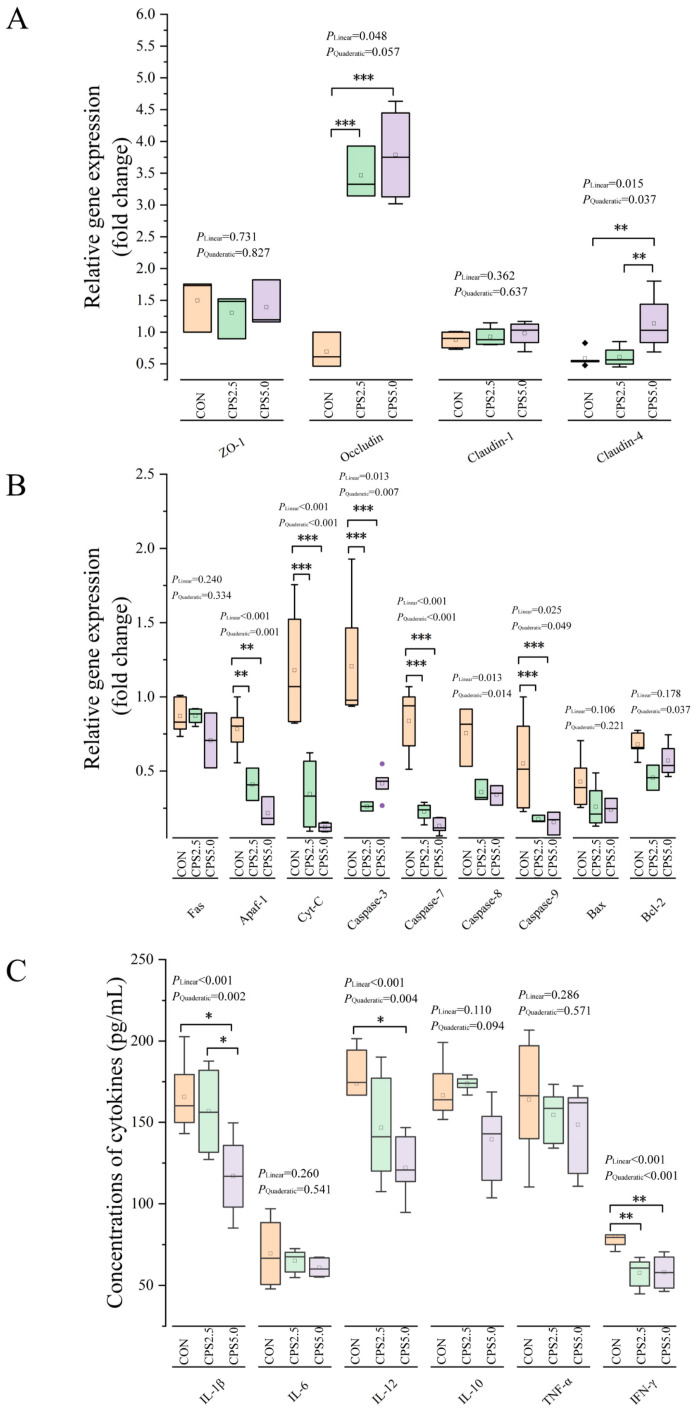
Boxplots of the mRNA relative expression of tight junction proteins and apoptosis-related genes and the concentrations of cytokines in the rumen epithelia of lambs supplemented with complex phytonutrients (*n* = 6). (**A**) The mRNA relative expression of tight junction proteins. (**B**) The mRNA relative expression of apoptosis-related genes. (**C**) The concentrations of cytokines. CON = control diet; CPS2.5 = supplemented with 2.5 g/kg complex phytonutrients; CPS5.0 = supplemented with 5.0 g/kg complex phytonutrients. * *p* < 0.05, ** *p* < 0.01, *** *p* < 0.001.

**Figure 2 animals-15-00228-f002:**
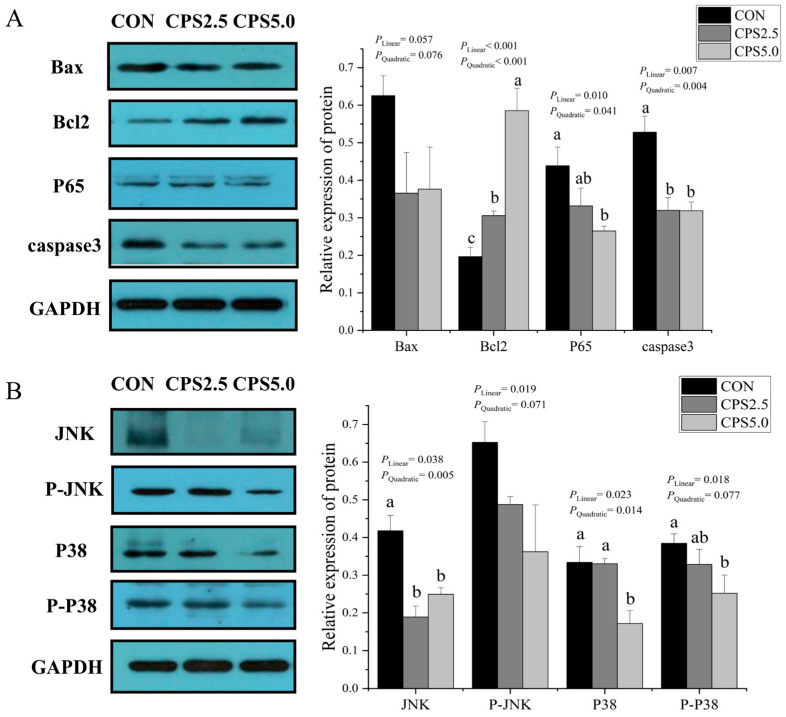
Effects of complex phytonutrient supplementation on the protein expression levels of JNK/P38 MAPK signaling pathways in the rumen epithelia of lambs (means ± SEMs; *n* = 6). (**A**) Protein levels of B-cell lymphoma-2-associated X protein (Bax), B-cell lymphoma-2 (Bcl-2), P65, and caspase3 in the rumen epithelia of lambs in the CON, CPS2.5, and CPS5.0 groups. (**B**) Phosphorylation levels of c-Jun N-terminal kinase (JNK) and p38 MAPK in the rumen epithelia of lambs in the CON, CPS2.5, and CPS5.0 groups. GAPDH represents the internal control for normalization. CON = control diet; CPS2.5 = supplemented with 2.5 g/kg complex phytonutrients; CPS5.0 = supplemented with 5.0 g/kg complex phytonutrients. All experiments were repeated more than three times. (Significant differences (*p* < 0.05) between groups indicated with different superscripts, a and b.) *n* = 6/group.

**Figure 3 animals-15-00228-f003:**
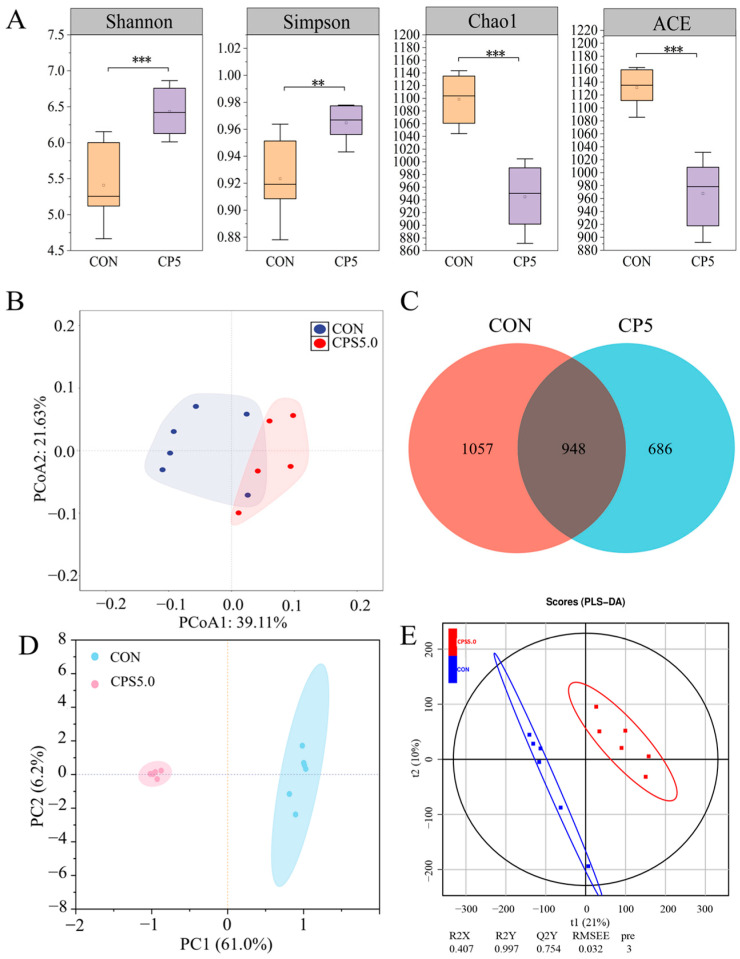
Effects of complex phytonutrient supplementation on rumen microbiota structure and diversity (*n* = 6). (**A**) Alpha diversity indexes. (**B**) Beta diversity analysis (PCoA plots). (**C**) Venn diagram. (**D**) The principal component analysis (PCA). The shaded areas denote the 95% confidence regions for two groups. (**E**) OPLS-DA plot. CON = control diet; CPS5.0 = supplemented with 5.0 g/kg complex phytonutrients. ** *p* < 0.01, *** *p* < 0.001.

**Figure 4 animals-15-00228-f004:**
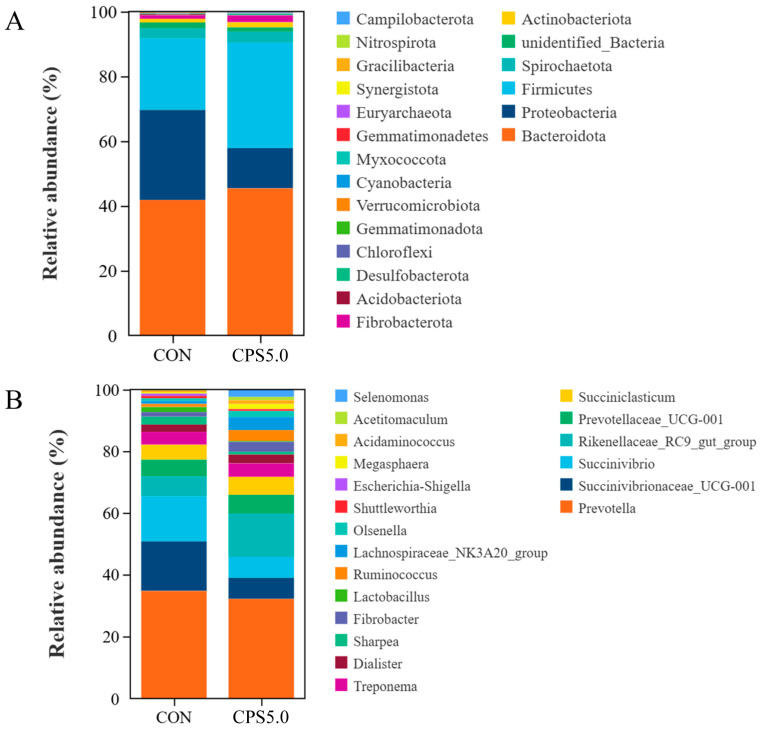
Effects of complex phytonutrient supplementation on ruminal bacteria at phylum and genus levels (*n* = 6). The relative abundances (%) of bacterial phyla (**A**) and genera (**B**) in the ruminal epithelium-associated microbiome of lambs fed CON and CPS5.0. CON = control diet; CPS5.0 = supplemented with 5.0 g/kg complex phytonutrients.

**Figure 5 animals-15-00228-f005:**
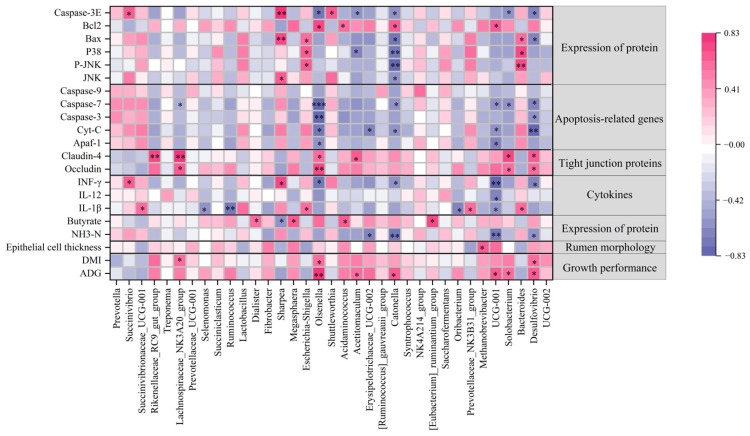
The heatmap of correlations between ruminal bacterial and rumen health-related indices. Positive and negative correlations are shown in pink and violet, respectively. * *p* < 0.05, ** *p* < 0.01, *** *p* < 0.001.

**Figure 6 animals-15-00228-f006:**
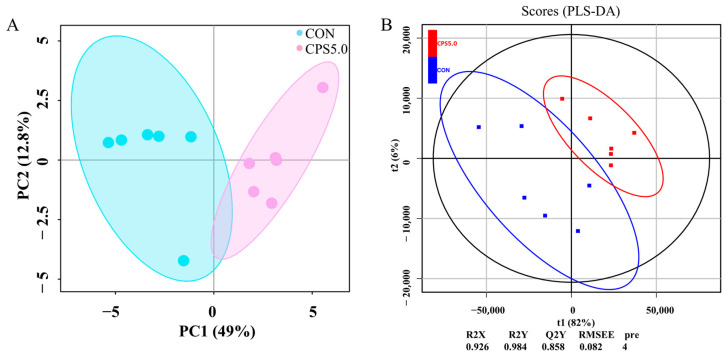
Metabolomic profiling of the ruminal contents of lambs fed CON and CPS5.0 diets (*n* = 6). (**A**) Principal component analysis (PCA) plot. The shaded areas denote the 95% confidence regions for the two groups. (**B**) Orthogonal partial least squares discriminant analysis (OPLS-DA) score plots. CON = control diet; CPS5.0 = supplemented with 5.0 g/kg complex phytonutrients.

**Figure 7 animals-15-00228-f007:**
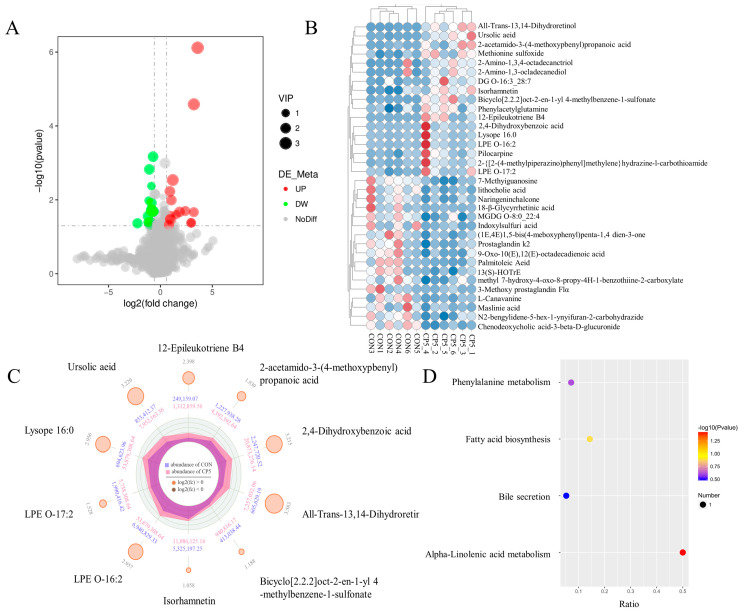
(**A**) Volcano plots. The upregulated and downregulated metabolites are shown in red and green, respectively, whereas gray dots represent the metabolites without significance in each group. (**B**) Heatmap of metabolites associated with the different treatments. Each column represents a sample, each row represents a metabolite, and the colors represent the relative contents of metabolites expressed in the groups; red shows that the metabolite was expressed at high levels and blue shows lower expression. (**C**) Radar map analysis of differential metabolites. Orange and brown circles: upregulated and downregulated metabolites; the size of the circles changes according to the log2 (FC) value; second circle: the outer circle represents the average expression levels of metabolites in the CPS5.0 group, while the inner circle represents the average expression levels of metabolites in the CON group; irregular shapes in the circles: the abundances of metabolites expressed in CPS5.0 and CON along each axis. (**D**) Differential metabolism KEGG pathways analysis. CON = control diet; CPS5.0 = supplemented with 5.0 g/kg complex phytonutrients; *n* = 6.

**Figure 8 animals-15-00228-f008:**
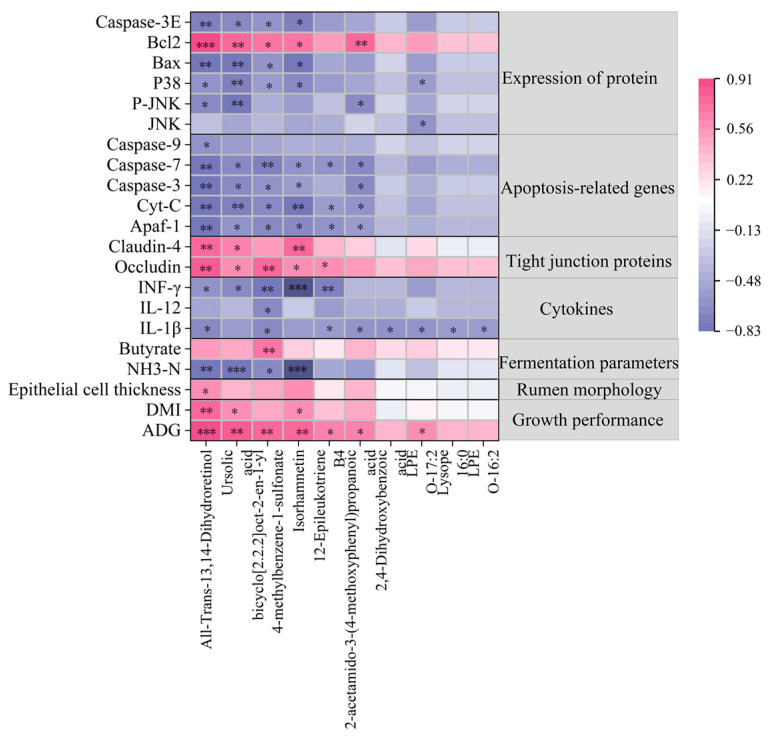
The heatmap of correlations between metabolite levels and rumen health-related indices. Positive and negative correlations are shown in pink and violet, respectively. * *p* < 0.05, ** *p* < 0.01, *** *p* < 0.001.

**Figure 9 animals-15-00228-f009:**
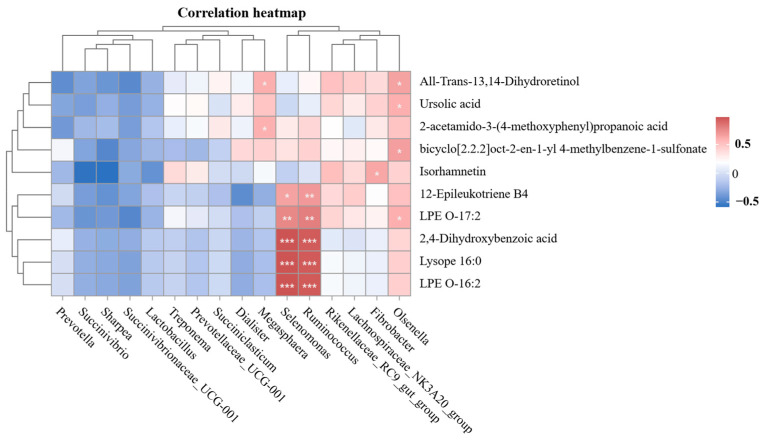
The heatmap of correlations between microbiome composition and metabolite levels. Positive and negative correlations are shown in red and blue, respectively. * *p* < 0.05, ** *p* < 0.01, *** *p* < 0.001.

**Table 1 animals-15-00228-t001:** Ingredients and nutrient levels of the basal diet (air-dry basis, %).

Items	Content
Ingredients	
Corn grain	35.1
Corn germ meal	23.0
Soybean meal	5.0
Cottonseed meal	5.0
Sunflower seed shells	13.0
Rice bran meal	12.0
Limestone	1.5
Salt	0.7
Vitamin–mineral premix ^1^	2.0
Dicalcium phosphate	0.7
Bentonite	2.0
Total	100.0
Nutrient levels ^2^	
Dry matter, %	91.8
Crude protein, %	16.6
Neutral detergent fiber, %	42.2
Acid detergent fiber, %	15.1
Ash, %	10.3
Calcium, %	1.05
Phosphorous, %	0.49
Metabolizable energy, MJ/kg^2^	9.63

^1^ Premix provided the following per kilogram of diet: vitamin A: 350,000 IU; vitamin D3: 93,750 IU; vitamin E: 0.938 g; vitamin K3: 0.063 g; vitamin B1: 0.062 g; vitamin B2: 0.188 g; niacin: 0.750 g; pantothenic acid: 0.500 g; vitamin B6: 0.062 g; biotin: 3.7 mg; folic acid: 0.038 g; vitamin B12: 0.7 mg; Se: 0.018 g; Zn: 3.000 g; I: 0.023 g; Co: 0.030 g; Mn: 2.500 g; Fe: 3.240 g; Cu: 0.500 g. ^2^ Metabolizable energy was a calculated value, while the others were measured values.

**Table 2 animals-15-00228-t002:** Effects of complex phytonutrient supplementation on the growth performance of lambs (*n* = 6).

Items ^1^	Groups ^2^			SEM ^3^	*p*-Value ^4^		
	CON	CPS2.5	CPS5.0		ANOVA	Linear	Quadratic
IBW, kg	30.04	30.64	30.45	0.639	0.932	0.783	0.923
FBW, kg	40.39 ^B^	45.80 ^A^	45.42 ^A^	0.705	0.040	0.016	0.006
ADG, g d ^−1^	196.3 ^B^	240.6 ^A^	242.9 ^A^	7.589	0.046	0.022	0.034
DMI, g d ^−1^	1321 ^B^	1626 ^A^	1591 ^A^	33.08	0.007	0.008	0.002
FCR	6.74	6.66	6.41	0.305	0.912	0.649	0.896

^1^ IBW = initial body weight; FBW = final body weight; ADG = average daily weight gain; DMI = dry matter intake; FCR = feed conversion ratio, DMI/ADG. ^2^ CON = control diet; CPS2.5 = supplemented with 2.5 g/kg complex phytonutrients; CPS5.0 = supplemented with 5.0 g/kg complex phytonutrients. ^3^ SEM = standard error of the mean. ^4^ ANOVA = contrast between CON, CPS2.5, and CPS5.0. (^A, B^ Within a row, values with different letter superscripts differ significantly at the *p* < 0.05 level.) Linear = linear effect of CPS addition; Quadratic = quadratic effect of CPS addition.

**Table 3 animals-15-00228-t003:** Effects of complex phytonutrient supplementation on the ruminal morphology of lambs (μm; *n* = 6).

Items	Groups ^1^			SEM ^2^	*p*-Value ^3^		
	CON	CPS2.5	CPS5.0		ANOVA	Linear	Quadratic
Papillae length	2155	2194	2379	149.2	0.818	0.268	0.512
Papillae width	470.5	494.1	501.7	1.35	0.766	0.401	0.689
Muscle layer thickness	2176	2372	2378	87.78	0.343	0.164	0.292
Epithelial cell thickness	127.6 ^B^	146.5 ^AB^	152.5 ^A^	7.35	0.042	0.019	0.057

^1^ CON = control diet; CPS2.5 = supplemented with 2.5 g/kg complex phytonutrients; CPS5.0 = supplemented with 5.0 g/kg complex phytonutrients. ^2^ SEM = standard error of the mean. ^3^ ANOVA = contrast between CON, CPS2.5, and CPS5.0. (^A, B^ Within a row, values with different letter superscripts differ significantly at the *p* < 0.05 level.) Linear = linear effect of CPS addition; Quadratic = quadratic effect of CPS addition.

**Table 4 animals-15-00228-t004:** Effects of complex phytonutrient supplementation on the ruminal fermentation parameters of lambs (*n* = 6).

Items ^1^	Groups ^2^			SEM ^3^	*p*-Value ^4^		
	CON	CPS2.5	CPS5.0		ANOVA	Linear	Quadratic
pH	5.18	5.35	5.28	0.124	0.873	0.410	0.434
NH_3_-N (mg/dL)	18.07 ^A^	17.70 ^AB^	10.20 ^B^	1.721	0.036	0.004	0.002
Total VFAs (mmol/L)	12.71	11.16	12.35	0.431	0.200	0.792	0.205
VFAs (% molar)							
Acetate	44.31	44.05	44.10	1.189	0.973	0.856	0.973
Propionate	38.33	37.52	37.07	0.702	0.261	0.100	0.259
Butyrate	10.40 ^B^	12.62 ^A^	12.45 ^A^	0.455	0.002	0.169	0.003
Isobutyrate	0.24	0.31	0.31	0.134	0.221	0.407	0.672
Valerate	3.33	3.33	3.44	0.448	0.925	0.813	0.964
Isovalerate	0.31	0.43	0.34	0.087	0.562	0.711	0.561
A/P	1.15	1.17	1.22	0.048	0.334	0.145	0.338

^1^ NH_3_-N = ammonia nitrogen; VFAs = volatile fatty acids. ^2^ CON = control diet; CPS2.5 = supplemented with 2.5 g/kg complex phytonutrients; CPS5.0 = supplemented with 5.0 g/kg complex phytonutrients. ^3^ SEM = standard error of the mean. ^4^ ANOVA = contrast between CON, CPS2.5, and CPS5.0. (^A, B^ Within a row, values with different letter superscripts differ significantly at the *p* < 0.05 level.) Linear = linear effect of CPS addition; Quadratic = quadratic effect of CPS addition.

## Data Availability

All data generated or analyzed during this study are included in this published article (and its Appendix A). Microbiome sequencing reads are available in the Sequence Read Archive (SRA) of NCBI under accession project number PRJNA1183985 (http://www.ncbi.nlm.nih.gov/bioproject/1183985, accessed on 10 November 2024).

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
