# Peer review of "Supplementation with Complex Phytonutrients Enhances Rumen Barrier Function and Growth Performance of Lambs by Regulating Rumen Microbiome and Metabolome"

_animals, 2025, doi:10.3390/ani15020228_

Round 1
Reviewer 1 Report
Comments and Suggestions for Authors
General
Respected authors
All opinion was emitted with all respect to the efforts of the authors for the preparation of the experiment and its report
General
The purpose of this study was to evaluate the supplementation of complex phytonutrient enhance rumen barrier function and growth performance of lambs by regulating rumen microbiome and metabolome. Variables measured were pH of ruminal fluid, rumen lipopolysaccharide (LPS), papillae morphology and inflammatory modulators. Justification of the experiment is clear and methodology and statistical procedures were appropriate to fully reach the objective raised. In general, the experiment was carried out in an acceptable manner and Discussion was addressed in a good way. However, there are weaknesses that must be corrected before it considered to publish.
The main concern is that several description aspects of materials and methods are missing, or not be enough specified.
Specific
L1: Capitalize each word of title (please revise the style and format). Be congruent “Complex phytonutrient” or “Complex phytonutrients”
Simple Summary
L14: The acronym “CP” is extensively used and recognized for describe “crude protein” Please use different acronym to define “Complex phytonutrients” in order to not confuse the readers.
L15: “discovery”? or “was observed that”? or “confirmed that”?
Abstract
Must be improved.
L27: Please indicate: 1) number of lambs used their average weight and number of replication /treatment. Mention the concentrate to forage ratio of the basal diet used.
Materials and Methods
L94: Description of feeding system are missing. Basal diet was offered ad libitum? Was offered one, twice daily? How the refusals were measured? Specify.
L95: Supplementation was “as fed-basis” or on a DM basis? Clarify.
L95: supplementation was g/kg of CP or g CP/kg diet? How the supplementation was performed? Uses a carrier and then mixed with the basal diet? Dressed the CP on diet at moment of feeding? How was? Please specify
L96: Use the symbol “%”
Table 1.
Remove the extra dot at the final of title
Corn grain cracked? Grounded?
L108: Only the energy was calculated? Therefore, chemical composition was determined in the laboratory? If yes, please include in mat and Methods the sampling procedure of feed and the procedures of laboratory used.
L114: How was the rumen samples was taken?
L125: Change the sub-heading as: Growth-performance estimation and laboratories procedures
L128: How DM was computed?
L129: Please describe as: Average daily weight gain (ADG) was computed by difference between the final live weight minus start live weight divided between 60 days.
L131: Currently, “feed efficiency” is computed as gain to feed ratio (gain/feed). Please compute feed efficiency in this manner (not as feed conversion)
L144: In serum? Please describe how was blood samples taken and the procedures used to do so (both for taking and conservation).
L147: Rumen tissue samples? Please describe how was rumen tissue samples taken and the procedures used to do so (both for taking and conservation).
L210 (Statistical analysis)
How were the lambs randomized into pens and into treatments? Were blocked by weight? Totally randomized? How do you balance the initial weight? Because the lambs were grouped into pens (3/pen) the experimental unit to performance analysis must be de pen? Was it analyzed in this way? Please specify this in the statistical description.
Table 2
Please indicate as a first row the number of repetition /treatment (6)
Table 4
Considering that lambs were fasted (at least 12-h) and diet contained 40%NDF, the ruminal fluid pH seems to be too low. Please check this data.
Please correct expression:
Total VFA (mmol/L) 12.71, 11.16, etc
VFA (% molar)
Acetate 44.31
Propionate 38.33
L488-500: It is important to note that in this particular case, supplemental Phytonutrients increased daily weight gain as direct result of greater DM intake, thus the improvement of ADG is 100% explained by increases on energy intake, but not to improvements in the utilization of dietary energy efficiency. This is confirmed with your data of rumen fermentation in which acetate to propionate ratio was not different between treatments. Therefore, the statement "supply additional energy for the growth of lambs" it shouldn't be the main argument to sustain your results. I recommend that you address your results as follows " Although other studies have shown that phytonutrients increase weight gain in sheep but also the efficiency of energy use (Arteaga-Wences et al., 2021; Estrada-Angulo et al. 2021;2014), in the current experiment, apparently, rumen fermentation was not the main via to improve growth, but the health of rumen could explain the improvements observed here" And then, the focus of your discussion on rumen health aspects and microbiome as is presented is timely and congruent.
L607: The NH3-N ruminal concentration could be mediated by feed N compounds degradation rate (results in high ruminal concentration) and by high NH3-N utilization (bacterial uptake) or NH3-N absorption (results in low ruminal concentration). It is important to note that ruminal samples were taken once time, thus dynamics of NH3-N formation/utilization cannot be visualized. If Provotella increased N degradation rate, other bacterial community must be present to utilize in a similar rate the NH3-N produced. Thus the increased Provetalla population does not explained per se, the low concentration of NH3-N in ruminal fluid. Please, deeply in this argument.
Conclusion
Ok!
Reviewer 2 Report
Comments and Suggestions for Authors
According to the manuscript in title of “Supplementation of Complex phytonutrient enhance rumen barrier function and growth performance of lambs by regulating rumen microbiome and metabolome”. There is a significant strategic foundation for the potential application of phytogenic compounds to control rumen fermentation and increase ruminant animals' productivity. The writing and organization are phenomenal. The paper, in my opinion, has to be improved, and the following points are detailed in the PDF file.

Reviewer 3 Report
Comments and Suggestions for Authors
Animals-3398044 investigated the effects of supplementing sheep diets with phytonutrient on performance, ruminal contents and health indicators. This is an interesting experiment and I hope it will be revised from a scientific perspective and made public.
General comment
The objectives, experimental design, and discussion are good, but the abstract and results need revision. I recommend that you reconsider how you present ANOVA and multiple comparison p-values in the results. Presenting results accurately is essential in any scientific paper.
Specific comments
L39-41 Is it really marked rise?
L223 I thought that there were some results that should have been shown as p-values for multiple comparisons, but instead showed p-values for ANOVA.
L323, 407, Figure 3E, 6B What does Q2Y indicate?
L365 The text shows results for some microorganisms. What criteria were used to select these microorganisms?
L390-391 Is this result correct? Does it match the Figure?
L392-393 Is this result correct? Does it match the Figure?
L404 Can the overlap be quantified?
L410-417 Are these results scientifically accurate?
L456-461 Are these results scientifically accurate?
L461-465 Does it show results accurately?
Reviewer 4 Report
Comments and Suggestions for Authors
The article addresses a critical challenge in contemporary ruminant farming: achieving a balance between high-concentrate diets and maintaining optimal rumen health. The innovative emphasis on the use of complex phytonutrients (CP) presents a timely and sustainable approach. The study provides a thorough assessment of CP supplementation, evaluating its impacts on growth performance, ruminal fermentation, epithelial barrier integrity, microbial diversity, and metabolomics. The application of advanced methodologies, including qPCR, Western blotting, and high-throughput sequencing, significantly strengthens the validity and reliability of the findings. Moreover, the discussion incorporates relevant literature to effectively contextualize the results. However, certain aspects of the manuscript require refinement:
1. Methodology: A clear justification for the selection of treatment dosages is needed.
2. Discussion: Dividing the discussion into subsections to improve the logical flow and coherence of the presented findings.
3. Conclusion: The conclusion would be strengthened by incorporating specific recommendations for future research directions.
Reviewer 5 Report
Comments and Suggestions for Authors
This experiment aims to investigate the effects of dietary complex phytonutrients (CP) supplementation on growth perfor mance lambs. The study also examines the impacts CP per profile rumen fermentation and epithelial barrier function, rumen microbiota and metabolomic profiles of lambs.
The objectives and hypothesis of this study is very clear, and the manuscript has scope for publication. However, the manuscript needs to be carefully revised in light of the following comments:
Simple Summary:
L 16: apoptosis ....specifically what ? of what cells?
Abstract:
L 44: .... apoptosis..... Apoptosis is a very nice word and catchy in the media but it concerns many cells, even erythrocytes. You can't use such a general formulation
Introduction :
The rumen, an essential and specialized organ in ruminants' diges tive systems, relies on its microbial community to transform dietary plant fibers and low quality proteins into volatile fatty acids (VFAs)- VFAs are formed mainly from fiber, I don't like the protein in the sentence.
Materials and Methods
L 91: ...Han sheep breeding farm located in Baotou City, Inner Mongolia, China.
L 92 : ...A total of fifty-four female crossbred lambs, ....
Where is the truth?
If it is a mix, what kind?
Characterize the breed
Results
The immunoglobulin level should be expressed in mg/L and listed in a table
Discussion
No discussion about immunity
Conclusions
apoptosis.... as above
.......suppress the activation of the JNK/P38 MAPK inflammatory pathway by altering the com ...... may inhibit because we are not sure
References : Editorial errors
Round 2
Reviewer 1 Report
Comments and Suggestions for Authors
Dear Authors
I have read the revised manuscript and appreciate the authors' consideration of my previous suggestions. Authors have covered all of my observations in an acceptable manner, in such a way that I have no further observations.
Author Response
Dear Editor and honored reviewers,
We would like to express our sincere gratitude for the thorough review and valuable feedback on our manuscript. We are pleased to learn that the reviewers have recognized the quality of our research and have recommended our manuscript for acceptance. It is a great honor to receive your recognition of our work. Once again, we deeply appreciate the time and effort that the reviewers and editorial team have invested in evaluating our work, and we are grateful for their support.
Sincerely yours,
Yuan Wang (corresponding author)
Reviewer 4 Report
Comments and Suggestions for Authors
The authors have addressed my previous comments well; however, I would like to point out a few clarifications regarding specific aspects of the manuscript:
Major Recommandations:
- Dry Matter Intake: The authors mention that feed leftovers were recorded daily throughout the study, but the dry matter content of these leftovers was not determined. Given that the dry matter content of feed can vary significantly from the feed distributed, it is unclear how the authors determined dry matter intake. I recommend that the authors provide more details on how they accounted for the dry matter of the leftover feed or clarify how dry matter intake was estimated if the leftover feed was not analyzed for its dry matter content in the methodology section.
- Metabolizable Energy Estimation: The authors present the metabolizable energy parameter in the results section. However, it is not mentioned in the methodology section. The methodology for estimating metabolizable energy should be added in the methodology section, along with the appropriate reference.
- Feed Conversion Ratios: The authors present the feed conversion ratios in the results section, but they do not provide details on how these were determined in the methodology section. It is not clear whether feed intake was measured by pens and average daily gain was measured individually. Additionally, the feed conversion ratios are not discussed in the discussion section. I recommend that the authors clarify how the feed conversion ratios were determined and include a discussion of these findings in the context of their study.
- Orthogonal Polynomial Analysis: The authors used orthogonal polynomial analysis to evaluate the linear and quadratic effects. However, the results of this analysis are only presented in the results section and are not discussed in the discussion section. The authors should obligatorily discuss these results, especially their implications and the precise significance in relation to the animals.
Minor Recommendations:
- In the simple summary, avoid the use of abbreviations, as per the journal's guidelines.
- In the section detailing the age of the animals, include the standard deviation (or state "0" if no variation is present).
- In Table 2, clarify that A and B within a row indicate values that differ significantly at the P < 0.05 level.
Reviewer 5 Report
Comments and Suggestions for Authors
The comments have been addressed, recomend to accept in the present form.
Author Response

(The authors gave the same response as above.)
